# Current-Induced Spin Photocurrent in GaAs at Room Temperature

**DOI:** 10.3390/s22010399

**Published:** 2022-01-05

**Authors:** Yang Zhang, Yu Liu, Xiao-Lan Xue, Xiao-Lin Zeng, Jing Wu, Li-Wei Shi, Yong-Hai Chen

**Affiliations:** 1School of Materials Science and Physics, China University of Mining and Technology, Xuzhou 221116, China; yangzhang@cumt.edu.cn (Y.Z.); xuexiaolan@cumt.edu.cn (X.-L.X.); liwei5450@cumt.edu.cn (L.-W.S.); 2Key Laboratory of Semiconductor Materials Science, Institute of Semiconductors, Chinese Academy of Sciences, Beijing Key Laboratory of Low Dimensional Semiconductor Materials and Devices, Beijing 100083, China; liuyu@semi.ac.cn (Y.L.); xiaolinzeng@semi.ac.cn (X.-L.Z.); wujing@semi.ac.cn (J.W.)

**Keywords:** spin, spin-photocurrent, current-induced spin polarization

## Abstract

Circularly polarized photocurrent, observed in p-doped bulk GaAs, varies nonlinearly with the applied bias voltage at room temperature. It has been explored that this phenomenon arises from the current-induced spin polarization in GaAs. In addition, we found that the current-induced spin polarization direction of p-doped bulk GaAs grown in the (001) direction lies in the sample plane and is perpendicular to the applied electric field, which is the same as that in GaAs quantum well. This research indicates that circularly polarized photocurrent is a new optical approach to investigate the current-induced spin polarization at room temperature.

## 1. Introduction

The manipulation of the spin degree of freedom by an electric field is at the core of semiconductor spintronics [1,2]. A charge current in a semiconductor structure can lead to either a homogeneous electron spin polarization (current-induced spin polarization (CISP)) or a spin accumulation transverse to the current direction by spin-dependent scattering (spin Hall effect) [3,4,5]. CISP was first predicted theoretically in the 1970s for a 2D electron gas (2DEG) with Rashba spin-orbit coupling (SOC) and has been observed in various materials such as heterostructures, quantum wells, and doped bulk materials [4,6,7,8,9]. Most studies employ Faraday or Kerr rotation and spin-polarized positron beams to characterize the CISP of materials but only a few strong spin-orbit coupling systems succeed in observing CISP at room temperature [10,11,12,13]. However, we observed the CISP in doped bulk GaAs at room temperature with the help of the circularly polarized photocurrent. The use of circularly polarized light for spin injection and regulation has been widely studied in many materials. The resulting effects of the circularly photogalvanic effect (CPGE), anomalous circularly photogalvanic effect (ACPGE), and photo-induced anomalous Hall effect (PAHE), etc., are widely reflected in various semiconductors and topological materials [14,15,16,17,18]. CPGE is traditionally gyrotropic optical media with strong spin-orbit coupling. As a consequence, the effect is ordinarily under the control of angular momentum selection rules of excitation with circularly polarized light [19]. ACPGE and PAHE are derived from the inverse spin Hall effect of spin-polarized photo-generated carrier diffusion and the drift under an electric field, respectively [16]. The circularly polarized photocurrent in the above effect, varying linearly with the applied bias voltage, can be self-driven without the external bias voltage [20,21,22], whereas the relationship between the circularly polarized photocurrent and the applied bias voltage stemming from the CISP is nonlinear. Here, we find that the circularly polarized photocurrent caused by CISP is squared with the applied bias voltage and defines this circularly polarized photocurrent due to CISP as the current-induced spin photocurrent.

This paper introduces a new method to investigate CISP based on circularly polarized photocurrent without the application of an external magnetic field. Through further experiments, we not only observed the CISP in the p-doped bulk GaAs, but also found that the CISP direction of bulk GaAs grown in the (001) direction is similar to the GaAs quantum well grown in the (001) direction, which lies in the sample plane and is perpendicular to the applied electric field direction [23].

## 2. Materials and Methods

Molecular Beam Epitaxy (MBE) technology was further developed on the basis of vacuum evaporation technology by Arthur and Cho of Bell Laboratories in the United States in the 1970s. It is also one of the most versatile thin film preparation techniques that can be adopted to epitaxially grow ultra-thin single-layer and multi-layer single-crystal films of semiconductors, metals, and insulators. MBE, also a special vacuum coating process, is a newly developed epitaxial film-forming method. As a new technology for preparing single-crystal thin films, epitaxy can grow thin films layer by layer along the crystal axis of the substrate material under the appropriate substrate and conditions. The advantages of this technology are as follows: First, the temperature of the substrate used is low; second, the film growth rate is slow; third, the beam intensity is easy to accurately control; and fourth, the film composition and doping concentration can be adjusted quickly with a change of the source. This technology makes it possible to prepare single-crystal films as thin as dozens of atomic layers, as well as ultra-thin quantum microstructure materials formed by alternately growing films with different compositions and different doping values.

MBE technology has the following characteristics:(1)The growth rate, equivalent to growing a single atomic layer per second, is extremely slow, about 1 µm/hour, which contributes to the precision of the control of thickness, structure, and composition, and the formation of steep heterostructures. In fact, as an atomic-level process technology, MBE is particularly suitable for growing superlattice materials.(2)The low temperature of epitaxial growth gives rise to the reduction of the lattice mismatch effect introduced by thermal expansion at the interface and the influence of self-doping and diffusion of substrate impurities on the epitaxial layer.(3)Since the growth is carried out in an ultra-high vacuum, the surface of the substrate can be completely clean after treatment, and contamination can be eliminated during the epitaxy process, so that an epitaxial layer of excellent quality can be shaped. In the molecular beam epitaxy device, there is generally an instrument to detect the structure of the surface, composition, and vacuum residual gas, which can also monitor the composition and structural integrity of the epitaxial layer at any time, thus conducive to scientific research.(4)MBE is a dynamic process, that is, the incident neutral particles (atoms or molecules) are stacked one by one on the substrate for growth, rather than a thermodynamic process. Therefore, it can grow thin films that are difficult to grow according to ordinary thermal equilibrium growth methods.(5)As an ultra-high vacuum (*p* < 10^−9^ Torr) physical deposition process, MBF requires neither the consideration of intermediate chemical reactions nor the influence of mass transmission, and the shutter can be employed to instantaneously control growth and interruption. Therefore, the composition and doping concentration of the film can be quickly adjusted with changes in the source.

A high-purity element source furnace is installed at the position facing the substrate in the MBE system, and the purity is generally above 99.9999%. Through the heating source furnace, the elements are sprayed onto the surface of the heated substrate at a certain thermal motion speed and according to a certain beam flow ratio, and they are piled up layer by layer to form a thin film. When using MBE epitaxial growth, the composition and doping level of the epitaxial layer hinge on the relative arrival rate of the component elements and dopants, and these parameters depend on the evaporation rate of the corresponding source.

This paper adopted MBE technology to grow high-quality p-type doped GaAs (the sample structure is as shown in Figure 1).

In polarization optics, the most commonly used polarization direction modulator is the magneto-optical modulator based on the Faraday effect. After linearly polarized light passes through the magneto-optical modulator, its polarization direction changes according to magnetic induction intensity, and then it is dynamically modulated by the current that generates the magnetic field. Magneto-optical modulators are now widely employed in the fields of biochemistry, pharmaceutical processing, sugar industry manufacturing, geometric measurement, and optical modulation technology, achieving highly precise measurement of the optical rotation of living substances, drugs, and sugars. Efforts are made to improve the angle measurement and the transmission accuracy of the azimuth, as well as the modulation method of the interferometer. Because the magneto-optical modulator modulates the current coil to generate the rise of temperature, the temperature of the magneto-optical modulator is considerably affected. At the same time, the magneto-optical modulator can only be used in the visible and infrared bands. Therefore, the application of the magneto-optical modulator has certain limitations, and it is imperative to find a new method of polarization direction modulation.

As a phase modulation device based on the photo-elastic effect, the photo-elastic modulator has the advantages of a large full field of view, low driving voltage, low power consumption, and a widely used band [6]. It is not only a high-precision polarization modulation and polarization detection technology, but also the key core device, especially in the field of polarization spectroscopy in the ultraviolet band, as it is currently the only practical polarization modulation device. In a photo-elastic modulator, a voltage-driven piezoelectric material (such as piezoelectric ceramics) applies a periodically varying mechanical force on an isotropic optical material (such as fused silica) to resonate the optical material to form periodic light. The elastic effect of the optical material produces periodic birefringence, generating the modulation of light of the phase delay after its passing through the photo-elastic modulator. The visible light elastic modulator itself does not generate heat, and does not have much impact on its temperature. In general, the peak delay of the photo-elastic modulator is adjusted to π/2 or π as a variable 1/4 wave plate or half wave plate. There are two basic modes of the use of photo-elastic modulators, namely the formation of the polarization modulator and the polarization state analysis device. The polarizer is inserted in front of the photo-elastic modulator and the transmission axis of the polarizer and the vibration axis of the photo-elastic modulator are set at an angle of 45° to form a polarization modulator. This paper utilizes this method to periodically generate circularly polarized light.

When measuring various physical quantities (such as temperature, acceleration, etc.), sensors are used to transform them into electrical signals, and then input to analyzers or measuring instruments. Sometimes, it is difficult to obtain solely the necessary signals. Normally, even unnecessary noise signals will be measured together. In fact, in various situations, various types of noises may be mixed up. In addition, on different occasions, sometimes the noise intensity is much higher than the required target signal level. If the measured signal is weaker, the noise will be relatively larger. The basic processing for weak signals usually helps to amplify the signal. The traditional amplification amplifies the noise and the signal simultaneously, without band-limiting or filtering processing, and any amplifying operation will give rise to the decline of the signal of the signal-to-noise ratios of the signal. Therefore, it is necessary to first use filtering to purify the signal and improve the signal-to-noise ratio in order to maintain the accuracy of the weak signals.

The lock-in amplifier is mainly a weak signal detection technology based on the cross-correlation method. Its core is the phase-sensitive detection technology using a reference signal with the same frequency and a fixed-phase relationship as the signal under test as a reference to filter out noises that are different from the reference frequency so as to extract useful signal components.

This paper uses the Stanford Research 830 lock-in amplifier to detect the ordinary photocurrent signal and corresponding circularly polarized photocurrent signal, respectively.

The geometry is illustrated in Figure 2. The four circle ohmic electrodes, whose radii are 0.25 mm with a distance of 2.5 mm, were made along the *y* and *x* directions, respectively, by indium deposition, and annealed at about 420 °C in a nitrogen atmosphere. The experimental setup is described as follows. A mode-locked Ti: sapphire laser with a repetition rate of 80 MHz serves as the radiation source. The incident light goes through a polarizer and a photo-elastic modulator (PEM), of which the peak retardation is set to be λ/4, to yield a modulated circularly polarized light with a fixed modulating frequency at 50 kHz (1F). An optical chopper with a rotation frequency of 223 Hz is used. The Gaussian profile light beam irradiates vertically on the sample with a diameter of about 2 mm at the perpendicular bisector of the two circle electrodes, and the power is 2 mW. The spin photocurrent and ordinary photocurrent are collected through the two circle electrodes by two lock-in amplifiers with the synchronization frequencies set to be 50 kHz and 223 Hz, respectively.

## 3. Results

We first carried out the photoluminescence spectrum and photocurrent spectrum test of the sample, and the specific experimental data are shown in Figure 3. From Figure 3, it can be inferred that the peak position of the photoluminescence spectrum is around 860 nm, consistent with the band edge of GaAs, and the peak position of the photocurrent spectrum is similar. In order to discuss our experiment further, an 860 nm laser is regarded as our excitation light source.

Figure 4 shows the ordinary photocurrent and circularly polarized photocurrent under different applied bias voltages. An 860 nm laser serves as the excitation source to achieve inter-band excitation of GaAs bulk materials. From Figure 4a, we can see that the amplitude of the ordinary photocurrent changes linearly with the varied applied bias voltages. However, the circularly polarized photocurrent, as shown in Figure 4b, changes squarely with the varied applied bias voltages. The experimental configuration at this time is as follows. The incident angle is 30° and the incident plane of the laser is perpendicular to the measuring electrodes. It is an interesting phenomenon that the ordinary photocurrent shows a different trend of the applied bias voltage from that of the circularly polarized photocurrent, which may be attributed to the effect of current-induced spin polarization.

Some previous theoretical and experimental studies have illuminated that in a doped GaAs bulk material or a low-dimensional system, the DC current in the plane leads to the unification of spin polarization of the system. In our experiments, the disparities between the left circularly polarized photocurrent and the right circularly polarized photocurrent can be measured by adjusting the phase delay of the PEM. By adjusting the phase delay of the PEM to be λ/4, we can measure the difference between the photocurrent induced by left- and right-circularly polarized light. Hence, the circularly polarized photocurrent in Figure 4b can be written as $\Delta J=J_{\sigma +}-J_{\sigma -}$. Due to the CISP effect, the applied current in the plane can result in spin polarization of carriers in the p-doped GaAs bulk material. There will be a difference in the absorption of left circularly polarized light and right circularly polarized light. Here we define the difference between the left circularly polarized photo-generated carrier and the right circularly polarized photo-generated carrier as Δn. From the previous research, we conclude that Δn∝E in the CISP effect where E is in the applied electric field. According to the differential forms of Ohm’s law, it can be obtained that $\Delta J=\Delta \sigma *E=\Delta ne\mu *E\propto E^{2}$. This explains why the circularly polarized photocurrent has a square relationship with the applied bias voltage and why the ordinary photocurrent changes linearly according to the applied bias voltage. Because the circularly polarized photocurrent is generated by CISP, here we define this circularly polarized photocurrent that has a square relationship with the applied bias voltage (due to CISP) as the current-induced spin photocurrent.

Next, we continue to discuss the direction of current-induced spin polarization in p-doped GaAs bulk materials. Some studies have indicated that for GaAs quantum wells grown in the (001) direction, the average current-induced spin polarization direction is in-plane. In order to further explore the direction of current-induced spin polarization for p-doped GaAs bulk materials, we have conducted experiments with varying incident angles and azimuths of excitation light. Figure 5a reflects the dependence of circularly polarized photocurrent on the incident angle when the incident plane of the laser is perpendicular to the electrode contact direction. We find that the circularly polarized photocurrent caused by CISP is nearly zero at a normal incidence. Then, as the incident angle increases, the circularly polarized photocurrent, which is linear with the incident angle, follows the same trend. It should be pointed out that we can rule out the possibility of CPGE based on material symmetry and previous research in which the circularly polarized photocurrent varies linearly with the applied voltage. Therefore, we believe that the current-induced spin polarization direction in the p-doped GaAs bulk material in the (001) direction is in the sample plane similar to the GaAs quantum well grown in the (001) direction. Figure 5b is dependent on the circularly polarized photocurrent due to CISP as a function of the sample azimuth angle. We define the azimuth angle as 0° when the incident plane of the laser is perpendicular to the connection of measuring electrodes. From Figure 5b, it can be seen that the amplitude of the circularly polarized photocurrent resulting from CISP varies with the azimuth angle, and the maximum and minimum values are 0 and 180 degrees with opposite signs, respectively. The circularly polarized photocurrent has an approximate 360-degree periodic relationship with the azimuth angle. Therefore, it can be inferred that the direction of the current-induced spin polarization in the p-doped GaAs bulk material grown in the (001) direction is perpendicular to the direction of the applied electric field. The schematic diagram of current-induced spin polarization in the doped GaAs bulk material is depicted in Figure 6c.

## 4. Discussion

Next, we discuss the mechanism of action of CISP in p-doped GaAs bulk materials. Under a stationary current flowing in the *x* direction, the electron distribution is asymmetric in the *k* space, owing to the acceleration in an electric field and momentum relaxation when the spin is degenerated, as shown in Figure 6a. It has been demonstrated that as long as spin-up and spin-down states are degenerated in *k*-space, the energy bands remain equally populated, and the current is not accompanied by spin orientation for bulk materials [24]. However, there exists a surface electric field in our p-doped GaAs bulk sample, as confirmed by our previous work [25]. As a consequence, the spin degeneracy is lifted due to the lack of inversion symmetry. Moreover, the body inversion asymmetry (BIA) in bulk GaAs cannot be ignored in that the relaxation processes, including spin flip, are different for the spin-up and spin-down carriers, and the quasi-momentum transfer from initial to final states is also different [25], resulting in spin polarization, as shown in Figure 6b. Our previous research shows that for doped GaAs, the surface electric field is present [26]. Therefore, the asymmetry caused by the surface electric field cannot be excluded. Moreover, some studies contend that the CISP in doped GaAs is derived from the extrinsic mechanism [27]. Therefore, to confirm the mechanism of CISP in GaAs materials, further experiments are indispensable.

Then, we conducted a variable temperature measurement experiment. Figure 7 is the result of the temperature experiment. Here we define circular polarization as the ratio of the circularly polarized photocurrent to the ordinary photocurrent. Through Figure 6, it can be concluded that circular polarization first rises and then drops with the temperature. In order to explain this phenomenon, a phenomenological discussion was conducted.

Phenomenologically, the average unbalanced spin polarization *S* of the electrons has the following relationship with the electron current *j*:(1)$$j_{\lambda }=\underset{\mu }{\sum }Q_{\lambda \mu }S_{\mu }$$
(2)$$S_{\alpha }=\underset{\gamma }{\sum }R_{\alpha \gamma }S_{\gamma }$$

Here, *Q* and *R* are the second-order pseudosensors. Equation (1) describes the spin-current effect, and Equation (2) describes its inverse effect (direct current induces electron spin polarization, CISP). The similarity between the two formulas is that both a pole vector and an axis vector are linearly coupled. Generally, the average spin polarization generated in Equation (2) has an arbitrary orientation relative to the electron current, and the angle between the spin polarization and current is determined by the Dresselhaus and Rashaba effects.

Numerous theoretical studies have proved that in the two-dimensional current-carrying system with Rashba spin-orbit coupling, the spin polarization component is perpendicular to the electric field exists [28,29,30]. With the weak field limit and short-range impurity scattering potential, the non-zero y-component spin polarization and longitudinal conductance can be expressed as [121]:(3)$$\langle S_{y}\rangle =\frac{em\alpha }{2\pi }\tau_{p}E_{x}$$
(4)$$\sigma_{xx}=\frac{e^{2}N}{m}\tau_{p}$$

Here, $\tau_{p}=\frac{m\mu }{e}$ is the momentum relaxation time, *N* is the electron concentration, *e* is the electron charge, *m* is the effective mass, and *α* is the Rashba constant. Therefore, the spin-orbit coupling strength of the system can be obtained by measuring the conductance and polarization:(5)$$\alpha =\left(\frac{2\pi eN}{m^{2}}\right)\frac{S_{y}}{\sigma_{xx}E_{x}}$$

Equation (5) also provides a method to measure the Rashba coefficient.

As shown by the red (triangular) curve in Figure 7a, the circularly polarized photocurrent first increases to the maximum value as the temperature rises, and then gradually dwindles. This reveals a variation of the CISP with temperature. Furthermore, $\langle S_{y}\rangle \propto △n\propto \sigma_{c}\propto j_{c}$, and combined with Equation (3), we can obtain:(6)$$j_{c}\propto \langle S_{y}\rangle =\frac{em\alpha }{2\pi }\tau_{p}E$$

Therefore, $j_{c}$ is proportional to the momentum relaxation time ($\tau_{p}$) and changes with temperature. It can be seen from Figure 5a that in the temperature range of 110 K < T < 140 K, $\tau_{p}$ increases with the increase in temperature, and when T > 140 K, $\tau_{p}$ decreases with the increase in temperature. Because of $\tau_{p}=\frac{m\mu }{e}$, the experimental results also reflect the temperature dependence of mobility. As the temperature climbs, *μ* first rises and then falls, indicating weak phonon scattering at low temperatures and ionized impurities scattering dominates, and when reaching a higher temperature, the lattice impurity scattering dominates again.

Assuming that the system is dominated by the Rashba mechanism, according to Equation (5), the relationship between the Rashba spin-orbit coupling coefficient *α* and the circularly polarized photocurrent can be obtained:(7)$$\alpha =\left(\frac{2\pi eN}{m^{2}}\right)\frac{S_{y}}{\sigma_{xx}E_{x}}\propto \left(\frac{2\pi eN}{m^{2}}\right)\frac{j_{c}}{j_{pc}}$$

It can be seen from Equation (7) that the temperature dependence of the circular polarization reflects the temperature dependence of the Rashba spin-orbit coupling coefficient *α*. The results of this experiment are consistent with the results of the spin photocurrent experiment in the literature [15], that is, *α* tends to augment first and then diminish with the increase in temperature.

## 5. Conclusions

In conclusion, a new approach is introduced to explore CISP based on circularly polarized photocurrent without applying an external magnetic field. By comparing the difference between the ordinary photocurrent and the circularly polarized photocurrent under the applied bias voltage, the phenomenon that the circularly polarized photocurrent is altered nonlinearly with the applied bias voltage is explained, and a basic phenomenological model is established. In addition, we suggest that the CISP direction of bulk GaAs grown in the (001) direction is similar to the GaAs quantum well grown in the (001) direction, both in the plane, and is perpendicular to the applied electric field direction. Next, there is still a need for a discussion about the CISP generation mechanism in the doped bulk GaAs. It is the mechanism of spin-flip scattering caused by the inversion asymmetry of the body, and the mechanism of extrinsic remains unclear.

## Figures and Tables

**Figure 1 sensors-22-00399-f001:**
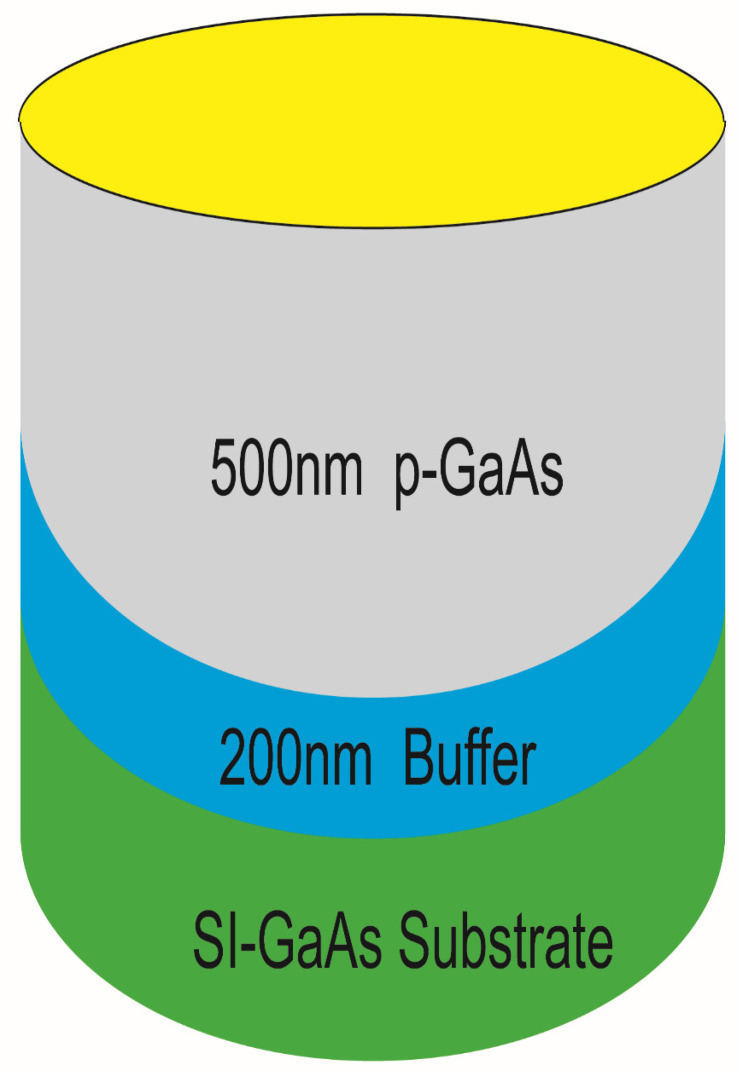
Schematic diagram of sample structure.

**Figure 2 sensors-22-00399-f002:**
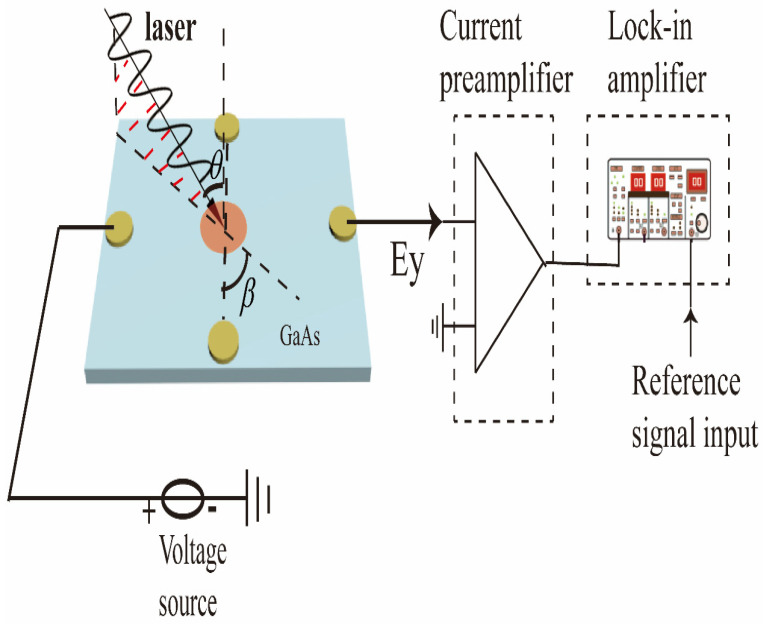
The schematic of the experimental setup. The golden circular patterns are the measuring electrodes and the red circular pattern represents the light spot. In the experiment, it defines the angle between the incident surface of the laser and the normal of the sample as the incident angle *θ*, and the angle between the incident plane of the laser and the line perpendicular to the measuring electrode is the azimuth angle *β*.

**Figure 3 sensors-22-00399-f003:**
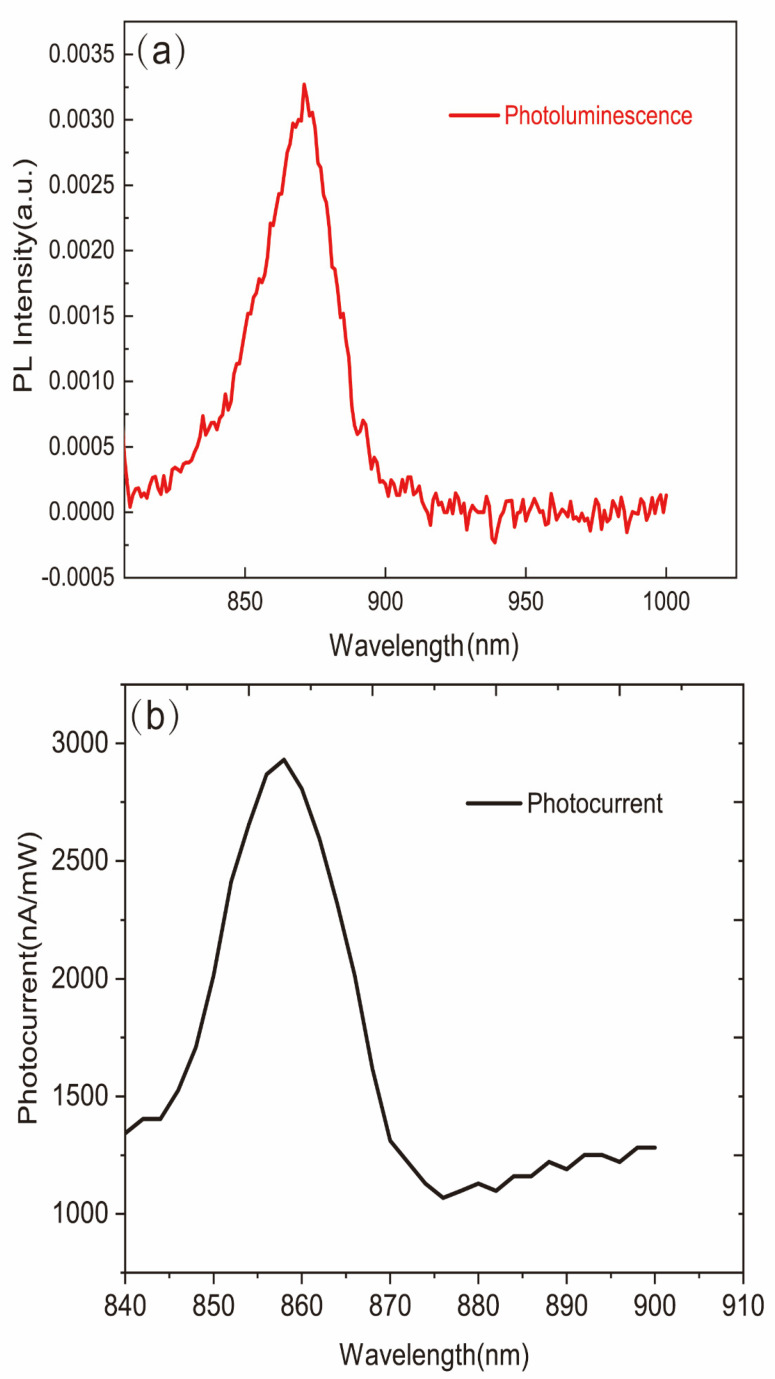
(**a**) The photoluminescence spectrum of p-GaAs; (**b**) the photocurrent spectrum of p-GaAs under 0.1 V.

**Figure 4 sensors-22-00399-f004:**
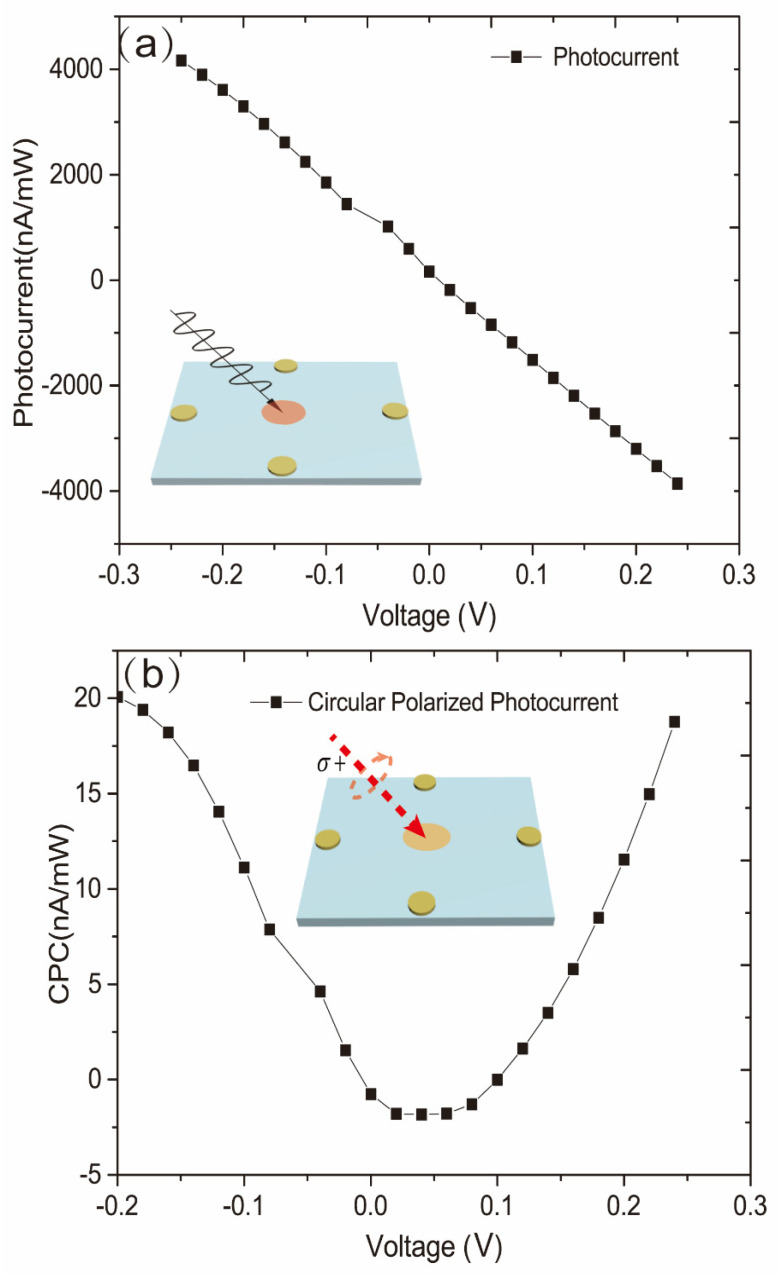
Dependence of (**a**) the ordinary photocurrent and (**b**) circularly polarized photocurrent on applied bias voltage when the incident angle is 30° and the incident plane of the laser is perpendicular to the measuring electrode.

**Figure 5 sensors-22-00399-f005:**
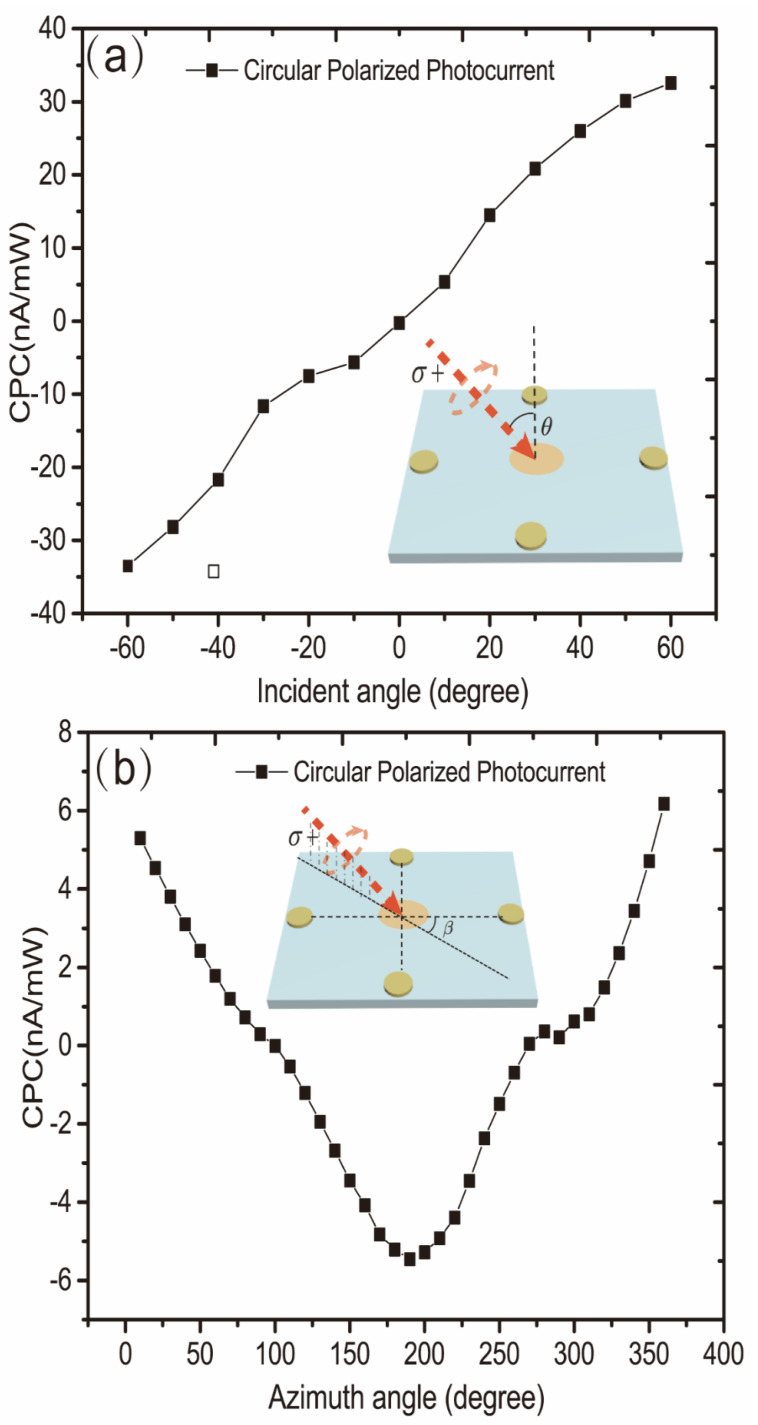
(**a**) The relationship between circularly polarized photocurrent and the incident angle when the azimuth angle is zero; (**b**) the relationship between circularly polarized photocurrent and the azimuth angle when the incident angle is 30°.

**Figure 6 sensors-22-00399-f006:**
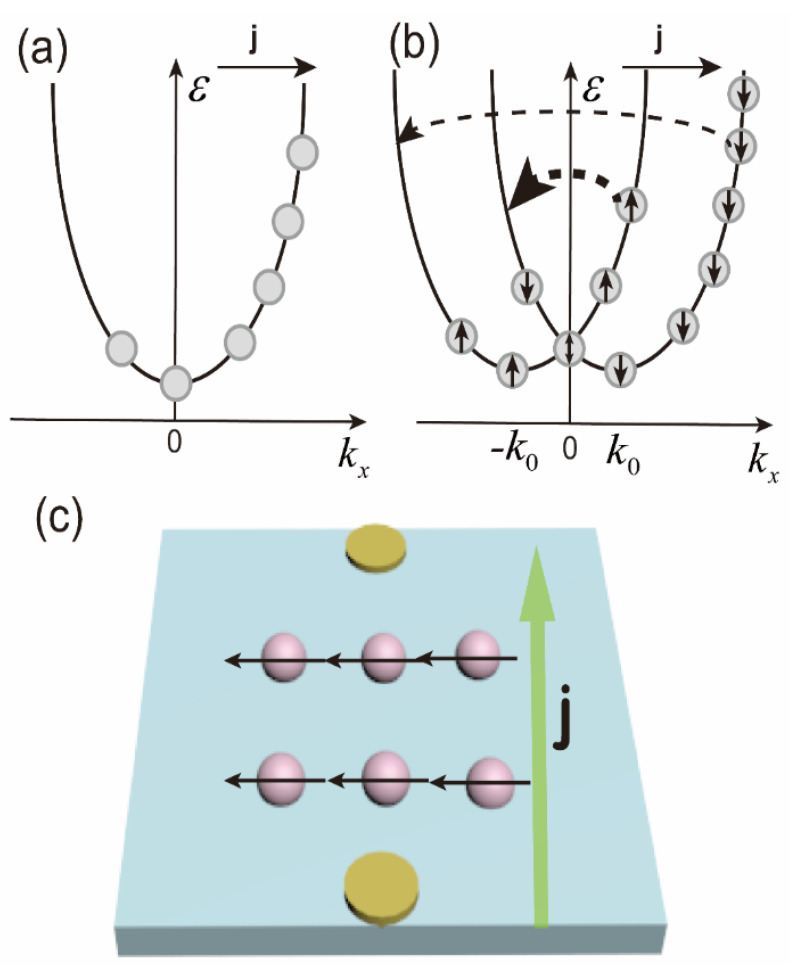
(**a**) Electron distribution at a stationary current flow stems from the acceleration in an electric field and momentum relaxation when the spin is degenerated. (**b**) Spin polarization is due to the spin-flip scattering. (**c**) The schematic of current-induced spin polarization. The pink sphere is photogenerated electrons, the green and black arrows are the electric field and spin polarization direction, respectively.

**Figure 7 sensors-22-00399-f007:**
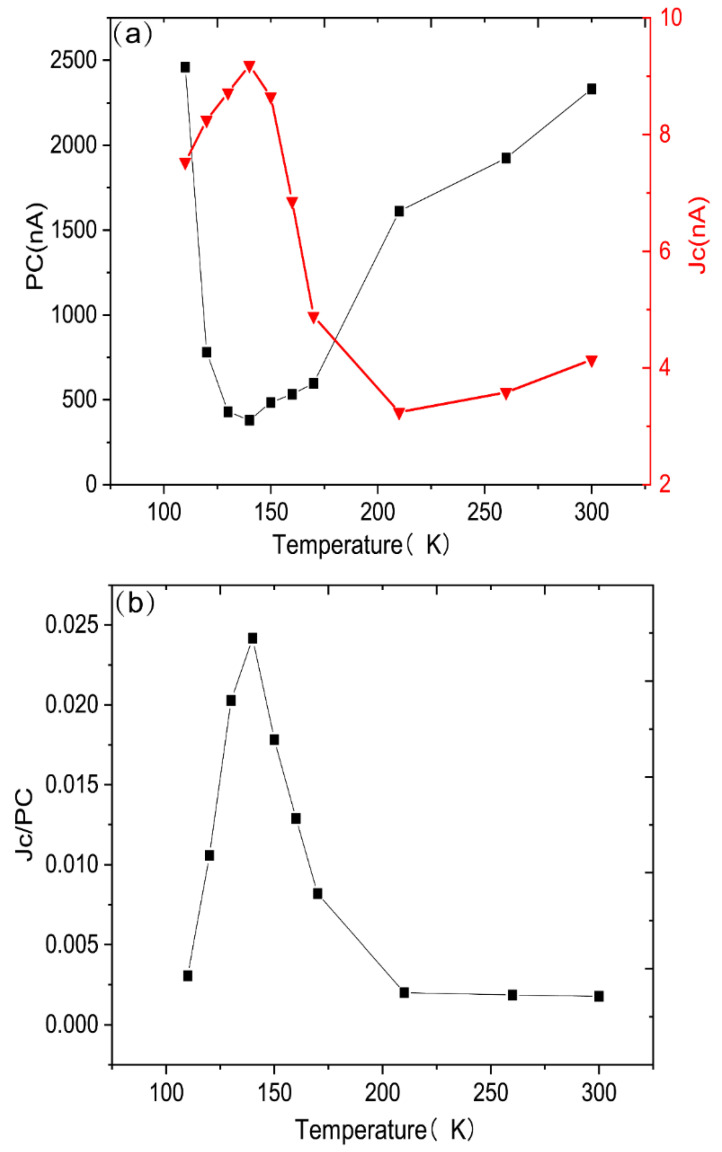
(**a**) The relationship between the photocurrent and the temperature. The black and red line are the ordinary photocurrent and circularly polarized photocurrent, respectively. (**b**) The relationship between the temperature and the ratio of circularly polarized photocurrent to ordinary photocurrent.

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
