# Peer review of "Current-Induced Spin Photocurrent in GaAs at Room Temperature"

_sensors, 2022, doi:10.3390/s22010399_

Round 1

Reviewer 1 Report

This manuscript proposes a method for studying CISP based on circularly polarized photocurrent without applying an external magnetic field. According this article, the circularly polarized photocurrent is observed in p-doped bulk GaAs and varies nonlinearly with the applied bias voltage at room temperature. In the current state, the work does not contain minimum requirements to be published as an original research work from my point of view. The detailed reasons are as follows.

  1. The introduction section should be revised and improved. The introduction does not introduce the reader with paper and a comprehensive literature review shall be conducted. It is expected that the introduction starts with the background and significance of the topic and then proceeds with a literature survey closely related to the topic, declaring research gaps, and finally announcing the contributions aiming the gaps.
  2. The comparative study with the similar researches is not considered. The presented results are not convincing.
  3. The quality of figures is not satisfactory.

Reviewer 2 Report

The paper fits the topics covered by the Journal, is well organized. It just require a proof-reading and improvement of Enlgish language usage.

Reviewer 3 Report

  1. It is not clear how to measure circular polarized photocurrent by reading the paper.

For example, in Figure 4(b), circularly polarized EM wave induced some current. It is called circularly polarized photocurrent. But in Figure 5, the incident EM wave is not circularly polarized as schematic, then the induced photocurrent is called circularly polarized photocurrent, too?

This part is very unclear. The authors should clearly define what is circularly polarized photocurrent in the paper. Does this mean photocurrent induced by circularly polarized EM wave? Is so, then the schematics of Figure 5 should be modified. If this is not the definition of circularly polarized photocurrent, then what is the definition? The authors should define "circularly polarized photocurrent" clearly in the paper.

  1. It is not clear from the measurement results of Figure 4 and Figure 5. What is the component of current induced spin photocurrent? Also, a more clear definition of "current induced spin photocurrent" should also be described in the paper.

Round 2

Reviewer 1 Report

The corrections carried out by the authors are acceptable.

Reviewer 3 Report

The authors have adequately responded to the reviewer's comments